# The Entorhinal Cortex and Adult Neurogenesis in Major Depression

**DOI:** 10.3390/ijms222111725

**Published:** 2021-10-29

**Authors:** Il Bin Kim, Seon-Cheol Park

**Affiliations:** 1Department of Psychiatry, Hanyang University Guri Hospital, Guri 11923, Korea; jonecaby49@gmail.com; 2Graduate School of Medical Science and Engineering, Korea Advanced Institute of Science and Technology (KAIST), Daejeon 34141, Korea; 3Department of Psychiatry, Hanyang University College of Medicine, Seoul 04763, Korea

**Keywords:** depression, entorhinal cortex, hippocampus, cognition, emotion, neural circuitry, neurogenesis

## Abstract

Depression is characterized by impairments in adult neurogenesis. Reduced hippocampal function, which is suggestive of neurogenesis impairments, is associated with depression-related phenotypes. As adult neurogenesis operates in an activity-dependent manner, disruption of hippocampal neurogenesis in depression may be a consequence of neural circuitry impairments. In particular, the entorhinal cortex is known to have a regulatory effect on the neural circuitry related to hippocampal function and adult neurogenesis. However, a comprehensive understanding of how disruption of the neural circuitry can lead to neurogenesis impairments in depression remains unclear with respect to the regulatory role of the entorhinal cortex. This review highlights recent findings suggesting neural circuitry-regulated neurogenesis, with a focus on the potential role of the entorhinal cortex in hippocampal neurogenesis in depression-related cognitive and emotional phenotypes. Taken together, these findings may provide a better understanding of the entorhinal cortex-regulated hippocampal neurogenesis model of depression.

## 1. Introduction

Globally, depression is one of the leading causes of disability [1]. The worldwide prevalence of depression is estimated to be up to 4.5%, and the total number of patients with depression has been reported to have increased by approximately 20% between 2005 and 2015 [2]. The prevalence of depression has remained stationary for several years, with high relapse and low remission rates [3,4,5,6]. Challenges in the psychiatric field have generated an emphasis on more personalized therapeutics based on endophenotype and biological markers and not just categorized diagnosis systems [7,8,9,10]. Inevitably, the current consequences of depression therapeutics demand a more comprehensive model of depression pathogenesis to allow for theoretical concepts that target partially remitted or refractory depression. For years, there have been requirements to refine the concepts of neural circuitry and neurogenesis, which are regarded as clinically targetable [11,12,13,14]. Indeed, brain networks and neural stem cell niches are known to have a major impact on depression-related phenotypes. The concepts of neural circuitry and neurogenesis may be more developed through an understanding of depression pathogenesis in terms of a comprehensive model concurrently implicating brain networks and stem cell niches. Thus, neural circuitry-regulated hippocampal neurogenesis may be a novel idea to improve our understanding of the pathogenesis of depression.

Defective hippocampal neurogenesis is a hallmark of depression. Ample evidence emphasizes a varying spectrum of neuropathology, from defects in hippocampal activity and volumes to alterations in activity-dependent gene expression, all of which suggest defective neurogenesis in the hippocampal dentate gyrus [15,16,17]. Given that hippocampal neurogenesis operates based on neural circuitry-mediated regulation [13,14,18,19,20], there have been tentative efforts to stimulate the upstream neural circuitry to enhance hippocampal neurogenesis in depression models. Interestingly, deep brain stimulation in animal depression models showed that entorhinal cortex stimulation improves cognitive performance via hippocampal neurogenesis, resulting in improvement in pattern separation and memory [21,22,23]. These findings may broaden the concept of entorhinal cortex-dependent hippocampal neurogenesis, which is also supported by a similar report on memory enhancement in human subjects [24]. Furthermore, recent ground-breaking studies using advanced approaches, which include chemogenetics, optogenetics, and molecular techniques, have also supported the role of the entorhinal–hippocampal circuitry and adult neurogenesis in the regulation of hippocampus-related cognition and emotion. For example, accurate stimulation of the glutamatergic afferent nerves from the entorhinal cortex was reported to result in the improvement in depression-related phenotypes in animal models of stress, which may be due to an enhanced hippocampal neurogenesis [25]. Accordingly, the entorhinal cortex-involved neural circuitry implicates hippocampal neurogenesis in cognition and emotion, both of which are frequently disrupted in depression. Therefore, the entorhinal cortex and hippocampal neurogenesis can be collectively exploited as a model to study neural circuitry and neurogenesis in depression. This review focuses on recent reports that support the biological association between the entorhinal–hippocampal circuitry and adult neurogenesis in the regulation of depression-related cognition and mood. Lastly, we suggest future directions for the entorhinal circuitry–hippocampal neurogenesis model of depression.

## 2. Entorhinal Cortex Involved in Regulation of Adult Neurogenesis

Our hypothetical concept of the entorhinal cortex-regulated hippocampal neurogenesis model of depression is based on both the neurobiological knowledge of adult neurogenesis and results of recent studies that not only demonstrate adult neurogenesis but also recapitulate the influence of neural circuitry and neurogenesis on depression-related phenotypes. We explored the neurobiological phenomenon of the co-operative engagement of the entorhinal cortex and hippocampus in adult neurogenesis [26,27,28,29], emphasizing a long-distance brain network in the modulation of neural stem cell niches in the hippocampus. Adult neurogenesis can modify the strength and number of synaptic innervations, which are regulated by dynamic mechanisms that provide synaptic stimulation in an activity-dependent manner to the neural stem cell niches in the hippocampus [19,20,30]. The entorhinal cortex is one of the major sources of excitatory input to the dentate gyrus of the hippocampus, which functions as a niche for the generation, maturation, and incorporation of granule cells that become part of the hippocampal circuitry. Stimulation of glutamatergic innervations in the hippocampus takes step-wise sequences in the regulation of adult hippocampal neurogenesis [31]. Early studies using patch-clamp electrical recordings showed that synaptic connectivity between entorhinal glutamatergic projections and granule cells develops at a neuronal age of 2 to 3 weeks after glutamatergic stimulation of the granule cells by the entorhinal cortex [32]. This is also consistent with another study that used a trans-synaptic tracing method [33]. Interestingly, the entorhinal cortex starts to develop glutamatergic innervations to adult-born neurons when the adult-born neurons reach 21–28 days of maturation [33]. This critical period indicates that the entorhinal cortex is involved in the regulation of adult hippocampal neurogenesis during the maturation phase. This is in line with a study that demonstrated this critical period, during which the stimulation of glutamatergic paths connecting the entorhinal cortex to the dentate gyrus facilitates an increase in the long-term potentiation of adult-born neurons [34,35]. From 28 to 42 days after birth, neurons are characterized by both a higher long-term potentiation amplitude and a lower threshold in response to physiological levels of stimulation. Although there are indirect modulations by which a non-cell-autonomous mechanism regulates the neural circuitry, these are beyond the scope of this review. Taken together, the entorhinal cortex regulates adult hippocampal neurogenesis in an activity-dependent manner (Figure 1). Therefore, the concept of entorhinal–hippocampal neural circuitry and adult neurogenesis is plausible. Considering this concept, we explored the question of whether the entorhinal hippocampal circuitry-neurogenesis model explains the pathogenesis of depression as previous studies have demonstrated that defective hippocampal neurogenesis is the hallmark of depression [36,37,38]. To corroborate this concept, we review up-to-date findings with an emphasis on entorhinal cortex-regulated hippocampal adult neurogenesis and depression-related phenotypes, including cognition and emotion.

## 3. Entorhinal Cortex-Regulated Hippocampal Neurogenesis for Cognitive Performance

Episodic memory deterioration is a key cognitive phenotype in depression [39,40,41,42,43,44,45,46]. Episodic memory damage is associated with volumetric reductions not only in the hippocampus [47,48,49,50], but also in the entorhinal cortex [51], suggesting that cognitive deterioration in depression may originate from neural pathogenesis involving the entorhinal cortex and hippocampus. In particular, mounting evidence indicates that defective hippocampal neurogenesis results in deteriorated episodic memory in depression [52,53,54,55]. One hypothesis is that loss of adult neurogenesis due to chronic stress may impair cognitive flexibility and pattern separation mediated by dentate gyrus contributing to symptoms of depression [14]. Nevertheless, the neurobiological mechanism by which the upstream neural circuitry affects episodic memory via hippocampal neurogenesis should be further explored.

The entorhinal cortex-hippocampal neural circuitry is regarded as the memory center of the mammalian brain and is mainly involved in the regulation of episodic memory, including object, spatial, and temporal information [56,57,58,59,60,61]. In humans, a recent study demonstrated that stimulation of entorhinal cortex facilitates improvements in memory- and learning-related processes. Deep brain stimulation of the entorhinal cortex leads to enhancements in spatial memory [24]. In a navigation task, human subjects receiving entorhinal stimulation reached a destination more quickly than controls without entorhinal stimulation. Entorhinal stimulation is accompanied by hippocampal theta rhythm resetting, which enables optimal induction of long-term potentiation, leading to fine coding of spatial information in the hippocampus [62]. In contrast, direct deep brain stimulation of the hippocampus does not change hippocampus-dependent memory performance [24,63], thereby underscoring the importance of targeting the upstream neural circuitry, and not just the hippocampus.

In addition, studies in mice have shown that deep brain stimulation of the entorhinal cortex enhances spatial memory and learning, which are accompanied by improved neurogenesis in the dentate gyrus (Figure 2a). In particular, Stone et al. demonstrated that high-frequency deep brain stimulation of the entorhinal cortex galvanizes neural stem cell niches to initiate consecutive neurogenesis processes, including dentate gyrus proliferation, immature progeny cell differentiation into adult-born neurons, survival of adult-born neurons for several (>5) weeks, and neuronal maturation into dentate granule cells [21]. Of note, the dentate granule cells are ultimately, but in a delayed fashion, incorporated into the existing hippocampal circuitry only after stimulation of the entorhinal cortex. Congruently, in the Morris navigation task, the spatial memory dependent upon the hippocampal circuitry is molded 6 weeks rather than 1 week after stimulation of the entorhinal cortex. This delayed effect of entorhinal stimulation is congruent with the maturation-dependent integration of adult-born dentate granule cells into the existing hippocampal circuitry, thereby leading to the modulation of spatial memory [64,65]. Researchers have also emphasized a causal relationship between entorhinal cortex-dependent hippocampal neurogenesis and spatial memory modulations by hindering neurogenesis and evaluating whether the spatial memory is improved. Other studies have also adopted a similar approach to understand the influence of entorhinal cortex stimulation on hippocampal neurogenesis and spatial memory [22,24].

In addition to deep brain stimulation, a preclinical study employed optogenetics to access the detailed mechanisms of memory regulated by the entorhinal–hippocampal circuitry. Robinson et al. addressed the question of whether the entorhinal–hippocampal circuitry modulates temporal memory, which is coded in the principal cells of the hippocampal CA1, also known as time cells [66]. They explored whether optogenetic inactivation of the medial entorhinal cortex leads to disturbances in hippocampal CA1 temporal coding, and thus time memory. The medial entorhinal cortex renders a major innervation to the hippocampus for regulating time as well as space information, while the lateral entorhinal cortex is involved in object information [67,68,69,70,71]. They applied bilateral optic fibers for light-induced inactivation of the medial entorhinal cortex while simultaneously recording from the hippocampal CA1 regions in rats injected with an adeno-associated viral vector in the medial entorhinal cortex. In a sequential object-treadmill-maze behavioral task, they assessed the influence of medial entorhinal inactivation on the hippocampal CA1 encoding activity for object, time, and space information in series. The rats were exposed to an object for an instant and then sent to a treadmill to run after a delay, followed by a second exposure to the object. Hence, temporal memory was assessed during the treadmill phase in a space-fixed setting. Notably, the inactivation of the medial entorhinal cortex triggered disturbance only in the CA1 time-encoding activity, but not in the object- and space-encoding activities. This result points to a distinctive mechanism of the entorhinal cortex-regulated hippocampal circuitry by which temporal experiences are integrated into episodic memory. Collectively, recent preclinical studies suggest that innervation from the entorhinal cortex is essential for hippocampus-dependent episodic memory. Nevertheless, the neural circuitry mechanism giving rise to memory dysregulation in depression is still in its infancy; thus, more studies using animal models of stress and biomolecular approaches are necessary to examine memory deficits in depression models based on the entorhinal cortex-regulated hippocampal neurogenesis.

Pattern separation defects are another hallmark of cognitive dysfunction in depression. The ability to distinguish between similar contexts is reliant on hippocampal neurogenesis [72,73,74,75,76]. Pattern separation defects are also regarded as biological markers of hippocampal neurogenesis disruption in depression [77,78]. Recent chemogenetic studies and neurogenesis ablation approaches collectively indicate the impact of neurogenesis defects on pattern separation deficits. Studies using functional brain imaging in combination with behavioral tasks further suggest that the entorhinal cortex is involved in pattern separation, which is also evidenced by neurobiological knowledge that the entorhinal cortex mediates communication between the hippocampus and neocortex to convey multiple cortical sensory and spatial information before sending it to the dentate gyrus, where existing and incoming contextual information of similar subjects are discerned using flexible encoding of varying activity patterns for different contexts [75,79]. Nevertheless, few studies have directly explored the relationship between entorhinal cortex-regulated hippocampal neurogenesis and pattern separation. Here, we review recent studies that investigated the imaging correlates of the entorhinal cortex for pattern separation, the effect of neurogenesis defects on pattern separation, and a recent chemogenetic method to clarify the causative relationship between entorhinal cortex-regulated hippocampal neurogenesis and pattern separation.

Functional brain imaging supports the idea that the entorhinal cortex is involved in the upstream neural circuitry of neurogenesis to drive pattern separation. Earlier studies using older human subjects examined the relationship between entorhinal circuitry defects and cognitive deficits, given that volumetric reductions in the medial temporal lobe, including the entorhinal cortex, have been reported to provoke cognitive deficits in both depression [80] and aging [81,82]. A recent study using high-resolution functional magnetic resonance imaging (fMRI) examined whether defects in the entorhinal cortex-regulated hippocampal neurogenesis result in defective pattern separation by gauging the functionalities of the entorhinal cortex, as well as the dentate gyrus and CA3 in the hippocampus [83]. In a context discrimination task, subjects with functional dissociation between the entorhinal cortex and hippocampus scarcely discriminated between similar objects, while their spatial discrimination remained undamaged. Specifically, the subjects demonstrated functional hypoactivity in the entorhinal cortex and, contrastingly, hyperactivity in the dentate gyrus and CA3, indicating that between-region functional activity imbalances may be associated with deficits in object discrimination. This is also in line with the neurophysiology that the lateral and medial entorhinal cortex are engaged in object and spatial discrimination, respectively [84]. This is further corroborated by a recent study that utilized rodents to show that the activities of lateral entorhinal neurons are directly associated with hippocampal CA3 activities, even though the human fMRI study provides indirect estimates of the neuronal activities from blood-oxygen level-dependent (BOLD) measurement [85]. Another fMRI study demonstrated that the entorhinal cortex may regulate pattern separation in older adults with or without a diagnosis of depression. They examined the activities of the amygdala, hippocampus, and lateral entorhinal cortex [86]. The patients with depression showed hypoactivity in the amygdala and hyperactivity in the entorhinal cortex and hippocampus during the object recognition task, wherein they exhibited deficits in discriminating between similar objects. The authors suggested that the entorhinal cortex may be engaged in the emotional modulation of pattern separation and that there may be an upstream circuitry, which includes the amygdala and entorhinal cortex, to control the hippocampus. Collectively, despite the fMRI experiments using older individuals just indicating the functional equivalents of pattern separation, the findings potentially provide evidence that the entorhinal cortex may be engaged in the hippocampal circuitry regulating pattern separation. Hence, further studies with different approaches are required to directly clarify the relationship between the entorhinal cortex-regulated hippocampal neurogenesis and pattern separation, and to carefully address whether hippocampal neurogenesis is a key mediator between neural circuitry and cognitive function.

Among the earliest studies exploring the impact of neurogenesis defects on pattern separation, X-ray irradiation of the hippocampus or synaptic plasticity disruption in dentate granule cells was mainly utilized to recapitulate neural stem cell niche defects to inspect impairment in the discrimination of a similar, safe context from a foot-shock context [78,87,88,89,90]. Specifically, Clelland et al. applied X-ray irradiation to the hippocampus to create a neurogenesis-ablated mouse model and scrutinized pattern separation defects by using both spatial discrimination and maze tasks [76]. The neurogenesis-ablated mice showed a diminished ability to detect subtle differences between similar contexts in both tasks. This is consistent with other studies in which neurogenesis-ablated mice exhibited impaired pattern separation in contextual fear conditioning tasks [78,90]. The neurogenesis-ablated mice presented similar freezing between a similar no-shock context and a shock-associated context, compared to controls that discriminate between the two contexts. These findings suggest that enhancing neurogenesis can improve pattern separation. Accordingly, Sahay et al. developed transgenic mice to selectively enhance adult neurogenesis [78]. In a fear conditioning task, transgenic mice with integrated adult-born neurons in the hippocampal dentate gyrus demonstrated significantly improved performance in the discrimination between similar contexts. Taken together, hippocampal neurogenesis ablation and genetic modification of neurogenesis approaches suggest that pattern separation is dependent upon hippocampal neurogenesis. Nonetheless, how the upper hippocampal circuitry regulates pattern separation remains poorly understood.

Recently, Yun et al. used a chemogenetic approach to generate transgenic mice with entorhinal cortex-targeted knockdown of a stress-provoked protein using an adeno-associated virus-mediated transfer gene (Figure 2b) [25]. Among the diverse stress-provoked proteins, they specifically studied *Trip8b*, whose knockdown excites hippocampal neurons, thereby enabling neurogenesis in the dentate gyrus. In the knockdown mice, entorhinal glutamatergic stimulation resulted in hippocampal neurogenesis, such as dendritic maturation of adult-born neurons by augmenting the inherent excitability of stellate cells in the entorhinal cortex. A contextual fear conditioning task was then implemented to inspect the pattern separation ability of the knockdown mice to distinguish a foot shock-related context from a similar context. The knockdown mice (*Trip8b*-shRNA) showed 50% more freezing behavior in the foot shock-related context than the control mice (SCR-shRNA), thereby demonstrating the improvement in pattern separation following entorhinal glutamatergic stimulation. In addition, X-ray irradiation of the dentate gyrus, leading to ablated neurogenesis, significantly diminished the effect elicited by entorhinal-targeted *Trip8b* knockdown. Taken together, the innovative work using a chemogenetic approach suggests that entorhinal cortex-regulated hippocampal neurogenesis regulates pattern separation. Nevertheless, the entorhinal–hippocampal circuitry-regulated neurogenesis mechanism underlying pattern separation still remains unclear; therefore, further preclinical research using brain stimulation and optogenetic or chemogenetic approaches to explain the entorhinal–hippocampal circuitry mechanism regulating pattern separation is required.

## 4. Entorhinal Cortex-Regulated Hippocampal Neurogenesis for Emotional Regulation

Hippocampal neurogenesis is associated with the inherent depression pathophysiology as well as the response to antidepressants [91,92]. Hippocampal neurogenesis defects are related to depression-related symptoms, such as hopelessness and helplessness [13,14,93,94]. Research indicates that defective neurogenesis alters a spectrum of physiological processes, including inflammation [95,96], the hypothalamic–pituitary–adrenal axis [97], and neurotrophic factors [98,99], which are all crucial to depression or stress responses. In addition, impaired neurogenesis lowers the effects of antidepressants, thus hindering recovery from depression [100]. Accordingly, increasing hippocampal neurogenesis is challenging, for which stimulation approaches have been applied to provoke behavioral effects in animal models of stress. In such approaches, the hippocampus is exposed to deep brain stimulation to enhance neurogenesis, which, however, does not alter hippocampus-dependent cognitive functions such as memory [24,63]. Rather, components of the upstream hippocampal circuitry, such as the entorhinal cortex, is a more suitable target for brain stimulation in improving depression-related symptoms, as research suggests that deep brain stimulation of the entorhinal cortex improves hippocampus-dependent memory [24,63]. Thus, the upstream hippocampal circuitry, including the entorhinal cortex, regulates neurogenesis and ameliorates depressive symptoms. Recently, a pioneering work revealed some clues regarding the relationship between entorhinal cortex-regulated hippocampal neurogenesis and depression-related phenotypes in animal models [25].

Currently, there are no available studies that have used superficial or deep brain stimulation aimed at the entorhinal cortex to link hippocampal neurogenesis and anti-depressive effects. Most studies have focused on other brain regions, including the prefrontal cortex [101,102,103,104,105,106,107,108,109,110,111,112,113,114], cingulate cortex [115,116,117,118,119,120,121,122], nucleus accumbens [123,124,125,126,127,128], thalamus [129,130,131,132,133], and striatum [134,135,136,137,138], to delineate defective neural circuitries that majorly contribute to the current depression circuitopathy [139]. Furthering the perspective of hippocampal neurogenesis can complement the depression circuitopathy by elucidating the influence of neural circuitry on neurogenesis and the relationship between neural circuitry-regulated neurogenesis and anti-depressive behaviors.

Molecular approaches, such as chemogenetic and optogenetic stimulation in combination with behavioral tasks may be an ideal option to unravel the relationship between entorhinal cortex-regulated hippocampal neurogenesis and anti-depressive behaviors. Yun et al. showed that chemogenetic stimulation of the entorhinal cortex ameliorates depression-related phenotypes in animal models of stress (Figure 2c) [25]. The authors designed an experiment to modulate hippocampal neuronal activity to increase neurogenesis by inducing maturation of dendritic morphology, and these hippocampal changes may result in anti-depressive behaviors. They used *Trip8b*, a specific stress-provoked protein that alters hippocampal neuron activity. Indeed, *Trip8b*-knockout mice demonstrated higher hippocampal neuronal firing frequency and increased neurogenesis with neuronal maturation than those in controls, mainly in the temporal dentate gyrus, which is known as a hippocampal region responsible for emotional processing and stress valance [140,141]. These results suggest that entorhinal cortex-targeted *Trip8b* knockdown facilitates hippocampal neurogenesis in an activity-dependent process that is regulated by the entorhinal cortex innervations to the dentate gyrus. To assess the behavioral effects of *Trip8b* knockdown mice, novelty-suppressed feeding and forced swimming tests were conducted in varying stress induction states, including basal state, acute stress state, and chronic stress state with varying exposures to corticosterone [142,143]. Under the different stress induction states, entorhinal cortex-targeted *Trip8b* knockdown facilitated anti-depressive behaviors that are shown both by an immediate response to feeding in the feeding test and increased mobility in the forced swimming test. The entorhinal cortex-regulated hippocampal neurogenesis was further investigated using a chemogenetic approach to examine whether glutamatergic or non-glutamatergic neuronal afferents are responsible for the anti-depressive effects. Yun et al. manufactured Gq-coupled human M3 muscarinic receptor-implanted mice that exhibited glutamatergic neurotransmission and mCherry-implanted mice that served as controls. They then showed the exclusive expression of CamKIIα-iCre-driven mCherry [144] in the entorhinal cortex and hippocampal dentate gyrus, demonstrating the accurate targeting of the entorhinal cortex-hippocampal dentate gyrus circuitry in both mouse models. They also showed more abundant c-Fos^+^ cells in the entorhinal cortex and hippocampal dentate gyrus in the M3 muscarinic receptor-implanted mouse than in the mCherry-implanted control mouse, indicating the enhanced glutamatergic neuronal activity elicited by designer ligand infusion of clozapine-N-oxide. Interestingly, chronic chemogenetic stimulation of entorhinal glutamatergic innervations to the dentate gyrus leads to anti-depressive behavior under basal and stress states. In the novelty-suppressed feeding test, the M3 muscarinic receptor-implanted mouse exhibited a 50% faster response to feeding than the mCherry-implanted mouse after an extended period (5 weeks), rather than a short period (3 weeks), of clozapine-N-oxide administration. In addition, the M3 muscarinic receptor-implanted mice spent more time interacting with a social target than the mCherry-implanted mice after exposure to chronic social defeat [145]. These results indicate that entorhinal glutamatergic afferents to the hippocampus modulate dentate gyrus neurogenesis, resulting in anti-depressive behaviors in animal models of depression. Taken together, preclinical work using a chemogenetic stimulation approach would further develop the field of depression circuitopathy, particularly that encompassing the entorhinal cortex and hippocampal dentate gyrus, which collectively contribute to the neural circuitry-regulated neurogenesis responsible for anti-depressive behaviors.

## 5. Suggestions for the Neural Circuitry-Neurogenesis Model of Depression

Pioneering preclinical studies examining entorhinal cortex-regulated hippocampal neurogenesis and cognitive and emotional symptoms of depression have led to initial advances in the neural circuitry–neurogenesis model of depression. Gomes-Leal et al. [13] suggested activation of reward centers for psychic well-being, for which good feelings are mediated in reward centers, such as nucleus accumbens and tegmental ventral area. Nevertheless, the mechanism of the depression model should be further elucidated in two parallel directions. First, future studies are required to explain the entorhinal cortex-regulated hippocampal neurogenesis in detail, with regard to the complex neurophysiology between entorhinal cortex and hippocampal dentate gyrus. During neurogenesis, the entorhinal cortex renders the main glutamatergic innervations to the dentate gyrus, by which progeny cells mature into granule cells that become incorporated into existing hippocampal circuitry [27,33,146,147,148]. Recent evidence suggests that the entorhinal cortex also renders GABAergic innervations to the hippocampus, which contribute to the entorhinal cortex-regulated hippocampal inhibitory circuitry that controls post-synaptic neuronal rhythmic theta activity in the dentate gyrus [149,150]. How excitatory and inhibitory entorhinal innervations regulate hippocampal neurogenesis and anti-depressive behaviors remain uncertain; therefore, further studies are required to elucidate this. Electrophysiological changes, such as gamma rhythm oscillations, elicited by theta rhythm alterations in the hippocampus, modify long-term potentiation in hippocampus-dependent cognition, including memory and learning [151]. The mechanism by which electrical rhythm is altered across the entorhinal cortex and hippocampus regulate neurogenesis and hippocampus-dependent emotions remains elusive. Anatomically, the entorhinal cortex has medial and lateral subdivisions that are well known to be engaged in the recognition of spatial and object representations, respectively [84]. This is consistent with recent research showing differential synaptic responses of the dentate gyrus to entorhinal subdivision innervations [152]. Responding to the medial entorhinal afferents, the adult-born granule cells excite the mature granule cells via N-methyl-D-aspartate receptors to build spatial representations. Responding to the lateral entorhinal afferents, adult-born granule cells inhibit mature granule cells via group II metabotropic glutamate receptors to build contextual representations. Accordingly, the question remains as to how the differential entorhinal subdivisions are associated with hippocampus-dependent cognitive performance as well as emotional regulation. In addition, hippocampal subdivisions related to emotional valances remain uncertain. For instance, the temporal hippocampal subregion regulating emotional processing [141] can be further examined with regard to the relationship between entorhinal cortex stimulation and hippocampal dentate gyrus neurogenesis that lead to anti-depressive behaviors. Therefore, the mechanism by which the entorhinal cortex is involved in the regulation of hippocampal neurogenesis and hippocampus-dependent cognitive and emotional functions can be further examined considering the multifarious perspectives of anatomical, neurobiological, and electrophysiological relationships between the entorhinal cortex and hippocampus.

Second, future studies are also required to scrutinize complex upstream hippocampal circuitries that include diverse brain subregions related to the pathogenesis of depression, while also considering hippocampal neurogenesis. Recent evidence suggests that various brain subregions, in addition to the entorhinal cortex, are also associated with dentate gyrus neurogenesis, indicating that complex neural circuitries may be implicated in hippocampus-dependent tasks as well [153,154]. For instance, high-frequency deep brain stimulation of the ventromedial prefrontal cortex results in upregulation of neurogenesis-associated genes and enhancement of neuronal proliferation in the hippocampus. The prefrontal-hippocampal circuitry is related to improved hippocampus-dependent object discrimination [155]. Furthermore, emotional memory circuitry has also been suggested in that basolateral amygdala activity regulates hippocampal neurogenesis, including fear context-related recruitment and proliferation of newborn neurons [156]. Stimulation of the anteromedial thalamus induces a 76% increase in the proliferation of progenitor cells in the hippocampal dentate gyrus [157]. Nevertheless, inconsistent findings have shown that the prefrontal cortex and nucleus accumbens scarcely support hippocampal neurogenesis [158]. The complex hippocampal circuitry is also regulated by multiple signaling neurotransmitters, including dopamine from the ventral tegmental area, acetylcholine from the diagonal band of Broca and septal nucleus, and serotonin from the median and dorsal raphe nuclei [19]. Investigations into the relationships between various neurotransmission signaling systems and depressive-related phenotypes also remain rudimentary. Thus, thorough research is required to delineate how the different brain subregions collectively contribute to the neural circuitry-regulated neurogenesis that regulates hippocampus-dependent cognitive and emotional functions.

## 6. Conclusions

Preclinical and clinical studies suggest that the upstream hippocampal circuitry may link the entorhinal cortex with adult neurogenesis, which in turn suggests that the neural circuitry neurogenesis mechanism can be a tenable concept to use when considering approaches to address defects in memory, pattern separation, and emotion, all of which are hampered in depression. In the treatment of depression, the neural stem cell niche in the hippocampus is considered an essential target of antidepressants, in addition to brain stimulation approaches to facilitate maturation of dentate gyrus neurons. Considering the link between hippocampal neurogenesis and antidepressant effects, the entorhinal–hippocampal circuitry is not only a valid neurophysiological concept, but also a novel area to explore in the pathogenesis of depression, with a particular emphasis on the potential role of the entorhinal cortex in regulating hippocampal neurogenesis, inducing improvements in memory, pattern separation, and anti-depressive behaviors. Therefore, entorhinal cortex-regulated hippocampal neurogenesis is a tenable example of the neural circuitry neurogenesis model that further expands our knowledge of depression pathogenesis related to cognitive and emotional symptoms in patients with depression, and ultimately aids in the development of novel therapeutic approaches.

## Figures and Tables

**Figure 1 ijms-22-11725-f001:**
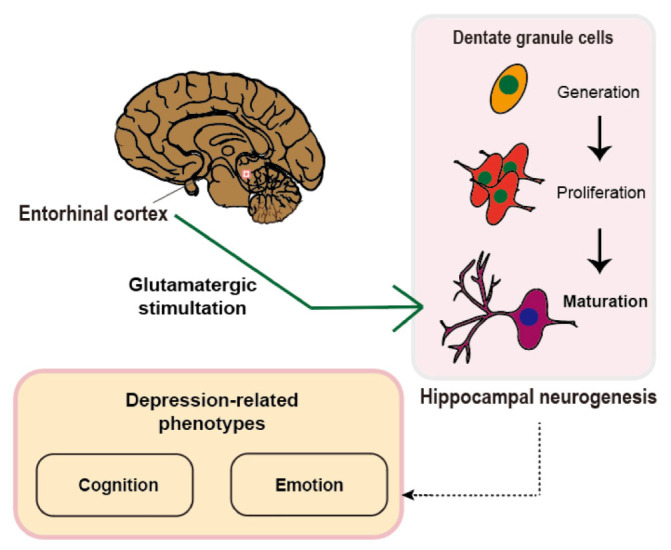
Hypothetical concept of the role of entorhinal cortex-regulated hippocampal neurogenesis in the manifestation of depression-related symptoms. The glutamatergic stimulation from the entorhinal cortex to the dentate gyrus of the hippocampus is illustrated. The glutamatergic stimulation regulates the maturation process of the dentate granule cells during hippocampal neurogenesis, which in turn affects cognition and emotion.

**Figure 2 ijms-22-11725-f002:**
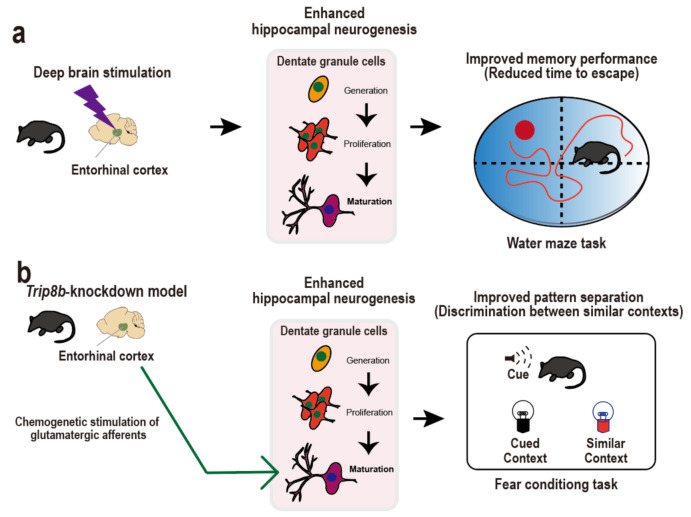
Supportive findings for entorhinal cortex-regulated hippocampal neurogenesis for cognition and emotion. (**a**), Entorhinal cortex-regulated hippocampal neurogenesis for memory function. Deep brain stimulation of the entorhinal cortex leads to increased neurogenesis, which presents as reduced time to escape in the water navigation maze task. (**b**), Entorhinal cortex-regulated hippocampal neurogenesis for pattern separation. Chemogenetic stimulation of the entorhinal cortex leads to enhanced neurogenesis in *Trip8b*-knockdown mice, which presents as improved discrimination between similar contexts in a fear conditioning task. (**c**), Entorhinal cortex-regulated hippocampal neurogenesis for emotion regulation. Chemogenetic stimulation of entorhinal cortex leads to improved neurogenesis in *Trip8b*-knockdown mice, which presents as an immediate response to a given food in the novelty-suppressed feeding task.

## Data Availability

Not applicable for the nature of the review study not involving humans or animals.

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
