# Peer review of "The Entorhinal Cortex and Adult Neurogenesis in Major Depression"

_ijms, 2021, doi:10.3390/ijms222111725_

Round 1

Reviewer 1 Report

This review focuses on the possible involvement of entorhinal –hippocampal circuitry dependent regulation of neurogenesis in depression. The paper is well-organized and supports a solid scientific hypothesis. Although the complexity of the depressive phenotype, this novel specific aspect related to the Entorhinal cortex function is well presented and clear. In general, the paper was a pleasure to read and I think will meet the interest of the readers. In particular, the novel findings that highlight a specific role for the entorhinal-hippcampal connections in the context of circuitopathies are relevant to better understand the pathogenesis of depression. Therefore, I recommend this paper for acceptance with only minor changes.

Thera are some errors throughout the text, I would suggest a carful revision.

Figure 1 and 2 need revision as the entorhinal cortex location is not correctly indicated either in the human (figure1) or rodent brain (fig.2)

Author Response

Response to Reviewer 1

Figure 1 and 2 need revision as the entorhinal cortex location is not correctly indicated either in the human (figure1) or rodent brain (fig.2)

  • According to your kind advice, we carefully adjusted the accurate location of entorhinal cortex in the figures.

    Response to Reviewer 1

    Figure 1 and 2 need revision as the entorhinal cortex location is not correctly indicated either in the human (figure1) or rodent brain (fig.2)

    • According to your kind advice, we carefully adjusted the accurate location of entorhinal cortex in the figures.

Reviewer 2 Report

GENERAL COMMENTS

The paper by Kim and Park (2021) reviews the recent literature on the implication of entorhinal cortex (EC) as an important area for regulating the hippocampal neurogenesis in the dentate gyrus, a well-established neurogenic niche in several mammalian species, including humans (Boldrini et al., 2018; Moreno-Jiménez et., 2019). Authors present several evidences that electrical stimulation of EC impacts adult hippocampal neurogenesis (AHN), which has been also implicated in affective disorders. The text is well written and recent literature is almost well covered, except for some important contribution dealing with AHN and affective disorders.

AHN is an important regulator of chronic stress, the main cause of depression.  The link between dentate gyrus and hypothalamus for regulating chronic stress is an important topic not discussed in the paper. Authors should check Anacker and Hen, 2017 (Nat. Rev. Neurosci. 18, 335–346. doi: 10.1038/nrn.2017.45) and Gomes-Leal, 2021 (Frontiers in Neuroscience Jun 16;15:594448. doi: 10.3389/fnins.2021.594448).  

Anacker and Hen review several aspects of how affective disorders, like anxiety and depression, can be a consequence of impairment of dentate gyrus function in the ventral rather than in dorsal hippocampuw. Kim and Park do not consider in their review different functions by ventral or dorsal hippocampus, although they discuss a lot on how EC stimulate adult neurogenesis in the dentate gyrus, mainly by deep brain stimulation. Authors should include information on that.

The paper by Gomes-Leal, 2021 updates information discussed by Anacker and Hen including an interesting discussing on how physical exercise is an important inductor of AHN and how brain evolution since Homo erectus, occurred in natural environments and with a lot of physical exercise. This was discussed in the context of neurogenic interventions like MAP-training (see papers by Tracey Shors) as important alternative therapies for affective disorders like depression, anxiety and post-traumatic stress disorders.

Gomes-Leal discusses the human correlate of an enriched environment, which had been established, liked motor stimulus, an important induction of AHN. Gomes-Leal states that “Considering that continuous aerobic exercise shaped brain evolution and developing, that is the likely explanation for the fact that running increases adult hippocampal neurogenesis (van Praag et al., 1999). It is possible that over the course of human evolution there was a kind of “neural symbiosis” between brain, physical activity and nature (natural landscapes), which might explain why aerobic exercise and enriched environment are so important for adult neurogenesis and cognition. This is supported by the fact that maturation of CNS circuits is dependent on sensory and motor stimuli (Chow et al., 1957; Narducci et al., 2018). Further experiments are needed to confirm this hypothesis. One such an experiment would be to investigate the effects of outdoor (parks with beautiful landscapes and lots of nice sensory stimuli) and indoor (ordinary fitness centers) aerobic physical activities in the mood of patients with depression and anxiety.

The link with the present paper is that Gomes-Leal proposed a neuroanatomical pathway linking sensory stimuli, AHN and reward centers. This authors states that “I would like to propose here that pleasant sensory stimuli contribute to psychological well-being through neural pathways connecting sensory cortices to neurogenic areas in DG and then to reward centers. There is a neuroanatomical basis for this hypothesis. Several sensory pathways project to entorhinal cortex, which send axons to hippocampal DG in several species, including rodents, primates and humans (Knierim et al., 2013;  Luna et al., 2019). From the ventral DG, there are axonalprojections to CA3 and then to reward centers, including nucleus accumbens (NA) in the ventral striatum of mice (Britt et al.,  2012; Bagot et al., 2015). The release of dopamine and serotonin in the reward centers contributes to hedonic feelings, including pleasure and happiness in humans (Worley, 2017; Loonen and Ivanova, 2019). There is a complex reciprocal interaction between new-born cells of the DG and neurons located in the lateral or medial part of the entorhinal cortex in mice (Luna et al., 2019).

Therefore, according to the proposed hypothesis, motor and sensory stimuli have profound effects on AHN, contributing to psychic well-being (Figure 1). This was likely shaped by our evolutive history.”

Kim and Park, must include those information in their review, as they cite a lot evidences that EC is fundamental of up regulation of AHN and this impact affective disorders. I think the inclusion of topics discussed by Anacker and Hen (2017) and the neural symbiosis proposed Gomes-Leal will strength the present manuscript.

SPECIFIC COMMENTS

Abstract

  1. Defects – It should be better impairment

Introduction

  1. Line 27. Reference 1. Include some new reference from WHO or other source
  2. Line 37. Cite recent contribution by Gomes-Leal, 2021. Frontiers in Neuroscience. Cite contribution by Anacker and Hen, 2017
  3. Line 47. Refs 14-16. Cite Anacker and Hen, 2017 ; Gomes-Leal, 2021. See above
  4. Lines 68-69. Authors must consider activation of reward centers for psychic well-being as proposed by Gomes-Leal 2021. Good feelings are mediated in reward centers, like nucleus accumbens and tegmental ventral area
  5. Line 109. Figure 1. The extremity of symbol in green normally means blockage. Please change to arrow or something like that
  6. Lines 121-123. “(..) Nevertheless, the neurobiological mechanism by which the upstream neural circuitry affects episodic memory via hippocampal neurogenesis remains unclear”. This mechanism has been proposed by Anacker and Hen, 2017. One hypothesis is that loss of adult neurogenesis due to d chronic stress may impair cognitive flexibility and pattern separation mediated by DG contributing to symptoms of anxiety and depression and other affective disorders.
  7. Line 281. Cite refs by Anacker and Hen, 2017 and Gomes-Leal, 2021
  8. Figure 2. The symbol used illustrate stimulation of DG by EC should be improved as the extremity means blockage. Please, including an arrow like structure
  9. Line 406. Reference 152 is considered a recent one. It is not.

In conclusion, as proposed by Gomes-Leal, natural sensory stimuli can be suitable for activating EC, which can in turn up regulate AHN decreasing stress and contribuing to happiness. Another alternative is deep brain stimulation as discussed by the authors.

Author Response

Response to Reviewer 2

Overall response:

  • We appreciate for your valuable advice for our manuscript. According to your kind suggestion for inclusion of the works by Gomes-Leal (2021) and Anacker and Hen (2017), we mentioned and cited those outstanding achievements in the revised manuscript.

Specific response:

Abstract

  1. We changed “Defects” to “impairments” in the abstract.

Main Text

  1. Reference 1
  • We cited more recent research (2017) that reported depression currents.
  1. Line 37. Reference addition
  • We added the citations of researches by Gomes-Leal and Anacker and Hen.
  1. Line 47. Reference addition
  • We added the citations of researches by Gomes-Leal and Anacker and Hen.
  1. Line 68-69. Contents revision
  • According to your valuable advice, we added the information indicating a reward center for psychic well-being in the suggestion sections in our revised manuscript.
  1. Line 109 Figure 1.
  • We changed the line shape using arrow.
  1. Line 121-123 Contents revision
  • According to your valuable advice, we added the information indicating the hypothesis that loss of adult neurogenesis due to d chronic stress may impair cognitive flexibility and pattern separation mediated by dentate gyrus contributing to symptoms of anxiety and depression and other affective disorders in our revised manuscript.
  1. Line 281 Reference addition
  • We added reference of research by Anacker and Hen (2017) and Gomes-Leal (2021).
  1. Figure 2.
  • We changed the line shape using arrow.
  1. Line 406 Reference addition
  • We added a recent research (2017) for the citation